

# Secure software development: leveraging application call graphs to detect security vulnerabilities

Lei Yan[1], Guanghuai Zhao[1], Xiaohui Li[1] and Pengxuan Sun[2]

[1] State Grid Beijing Electric Power Company, Beijing, China
[2] The Faculty of Information Technology, Beijing University of Technology, Beijing, China

## ABSTRACT

The inconsistency in software development standards frequently leads to vulnerabilities that can jeopardize an application's cryptographic integrity. This situation can result in incomplete or flawed encryption processes. Vulnerabilities may manifest as missing, bypassed, or improperly executed encryption functions or the absence of critical cryptographic mechanisms, which eventually weaken security goals. This article introduces a thorough method for detecting vulnerabilities using dynamic and static analysis, focusing on a cryptographic function dominance tree. This strategy systematically minimizes the likelihood of integrity breaches in cryptographic applications. A layered and modular model is developed to maintain integrity by mapping the entire flow of cryptographic function calls across various components. The cryptographic function call graph and dominance tree are extracted and subsequently analyzed using an integrated dynamic and static technique. The extracted information undergoes strict evaluation against the anticipated function call sequence in the relevant cryptographic module to identify and localize potential security issues. Experimental findings demonstrate that the proposed method considerably enhances the accuracy and comprehensiveness of vulnerability detection in cryptographic applications, improving implementation security and resilience against misuse vulnerabilities.

## INTRODUCTION

Cybersecurity has become a significant issue in today's information-driven society, especially within software development, where flaws in encryption methods can jeopardize sensitive information. Cryptography protects data security in the information exchange process, enabling essential functions like data encryption, key exchange, and authentication (*Chen et al., 2023*). Nevertheless, the reliability of cryptographic systems is not only based on theoretical security; it also hinges on how securely these systems are implemented in software (*Aldosary & Tanveer, 2024*). This distinction is essential: theoretical security pertains to the design and principles of cryptographic algorithms, whereas implementation security emphasizes how these principles are applied in real-world coding. Even if the theoretical design is sound, inadequate implementation can

Corresponding author
Pengxuan Sun,
SunPXBJUT@163.com

introduce vulnerabilities, affecting information systems' confidentiality, integrity, and authenticity (*Limkar et al., 2023*).

A rising concern in cryptographic software security is the issue of "cryptographic misuse vulnerabilities." These vulnerabilities arise when developers inadvertently breach cryptographic security protocols, often due to a lack of standardization or insufficient awareness of secure coding principles (*Hasan et al., 2024*). Misuse can manifest in various ways, including incorrect configuration, poor key storage practices, or insecure handling of cryptographic functions, leading to major security risks for end users. In contrast to design flaws in the algorithms, cryptographic misuse vulnerabilities are more varied and can have extensive consequences, making swift analysis and detection essential for maintaining software security (*Nyangaresi, 2023*).

Despite the significance of this domain, the current research on detecting cryptographic misuse vulnerabilities is still in its infancy and faces numerous challenges (*Baho & Abawajy, 2023*). For instance, existing analyses often do not clearly categorize misuse mechanisms, which complicates the comprehensive detection of vulnerabilities (*Alkhwaja et al., 2023*). Moreover, current automated detection methods usually only address a limited number of scenarios or depend on simple detection rules, resulting in low detection rates and excessive false positives (*Pimenta Rodrigues et al., 2024*). While manual analysis can be effective, it requires significant expertise and is not feasible for analyzing large amounts of software, thereby stressing needing more efficient and automated detection solutions.

This section reveals a new mechanism for identifying cryptographic misuse vulnerabilities in software applications. Unlike traditional vulnerability detection approaches that typically depend on static or dynamic analysis, our method integrates both to address the shortcomings identified in earlier techniques. For example, methods that rely solely on static analysis may overlook runtime vulnerabilities due to restricted path coverage. In contrast, those employing only dynamic analysis might miss implicit API calls or dependencies that were not executed during testing.

Our method closes these gaps by creating a comprehensive cryptographic function call graph (CFCG) through control flow reconstruction and employing a cryptographic function dominance tree (CFDT) to capture the interaction between function calls in a modular, layered format. This layered framework, which merges dynamic and static elements, offers more extensive coverage of cryptographic function flows, allowing our system to identify misuse vulnerabilities that previous tools may have missed. Moreover, our model minimizes false positives by refining detection rules customized to specific cryptographic modules, thereby improving the detection of misuse vulnerabilities without inundating analysts with unnecessary alerts.

The main contributions of this article are as follows:

- We propose an innovative, combined dynamic and static analysis approach for detecting cryptographic misuse vulnerabilities, using cryptographic function dominance trees to enhance accuracy and scope.

- We develop a layered, modular application integrity model that offers an in-depth perspective on cryptographic function flows and enables more effective detection of integrity-related flaws.
- We create targeted detection rules for addressing cryptographic misuse vulnerabilities in Windows applications, resulting in practical enhancements to software security analysis.
- We evaluate the effectiveness of our proposed method through a series of experiments, displaying its robustness in identifying critical misuse vulnerabilities within cryptographic applications.

This article is organized as follows: "Related Work" reviews relevant literature, analyzing existing methodologies and their shortcomings in vulnerability detection. In "Scheme Architecture: Proposed approach", we introduce the suggested detection method, providing details about the layered application integrity model and the role of the cryptographic function dominance tree in pinpointing misuse vulnerabilities. "Experimental Analysis" describes the experimental setup and evaluates the results of our approach, emphasizing its strengths and practical uses. Finally, "Conclusions and Future Work" concludes the article by discussing the key findings, potential limitations, and pathways for future research.

# RELATED WORK

Data breaches and vulnerabilities related to cryptographic misuse are ongoing concerns in the field of cybersecurity. Unauthorized access to personal, health, or financial information can pose serious privacy risks. A data breach occurs when sensitive information is revealed to individuals who are not authorized to see it, thereby compromising the data's privacy and integrity (*Pimenta Rodrigues et al., 2024*). Wearable devices, commonly used for everyday convenience, are a prime example of this risk. Their connectivity can make personal data vulnerable to potential cyberattacks that target communication channels (*Silva-Trujillo et al., 2023*).

## Data breach prevention and privacy-enhancing technologies

Researchers have created various frameworks and encryption schemes to address vulnerability in data communication. *Coutinho et al. (2018)* introduced an advanced neural cryptographic model that trains artificial neural networks to establish a theoretically unbreakable one-time pad (OTP) encryption algorithm. This model operates autonomously without needing human input to enhance communication security. *Vivar et al. (2020)* comprehensively analyzed security issues in smart contracts and recent advancements in available public tools. Meanwhile, *Bai et al. (2024)* proposed a password-based access control system that maintains security while ensuring system availability. *Xu et al. (2024)* developed the Advanced Network-Hiding Access Control (AHAC) framework, which minimizes exposure in network settings, thereby improving secure data access. *Jiang et al. (2024)* introduced an innovative Reversible Data Hiding in Encrypted Domains (RDH-ED) scheme based on Variable Threshold Image Secret Sharing (VTSIS),

which requires no preprocessing and supports high security, a high embedding rate, and complete reversibility.

*Valera-Rodriguez, Manzanares-Lopez & Cano (2024)* examined Fully Homomorphic Encryption (FHE) using the Microsoft SEAL library, tackling privacy and security challenges across digital ecosystems. *Jin et al. (2024)* designed a token-based encryption scheme that optimizes search and update times, supports multi-user modes, reduces memory usage, and addresses history storage challenges. *Chen et al. (2024)* proposed an ECC-based AKA protocol for secure message exchanges between devices and servers, ensuring user anonymity, forward security, and two-way authentication with minimal computational overhead.

## Detection of password misuse vulnerabilities

Password misuse continues to be a major vulnerability due to possible mistakes by developers in following cryptographic security standards. *Zheng et al. (2024)* tackled this issue by introducing CRYPTOLINT, a lightweight vulnerability detection system that employs static program slicing techniques to identify cryptographic misuse. *Shuai et al. (2014)* created the Cryptographic Misuse Analyzer (CMA), which uses static analysis to identify password API calls in Android applications. This system constructs control flow and function call graphs, performs dynamic analysis, and logs cryptographic function invocations to detect possible password misuse vulnerabilities.

To focus on cryptographic misuse in iOS applications, *Li et al. (2014)* developed iCryptoTracer. This system integrates static and dynamic analysis to identify algorithmic issues and offer detailed cryptographic insights.

## Vulnerability detection tools and techniques

Multiple tools have been established to address vulnerabilities more comprehensively, using automated testing methods to ensure strong security against diverse cryptographic misuse risks. The POET system (*Rizzo & Duong, 2010*) employs fuzzing technology to automate the detection of Padding Oracle attacks in web applications. At the same time, FIAT (*Barenghi et al., 2012*) uses fault injection techniques to improve security testing on password-based systems. Also, *Aumasson & Romailler (2017)*, *Peng et al. (2019)*, *Feng et al. (2022)*, *Kamalbayev et al. (2021)* proposed differential fuzzing technology, which uses the CDF automated tool to assess cryptographic applications for security vulnerabilities.

In summary, our proposed detection method employs a layered and modular integrity model, combined with dynamic and static analysis techniques, to create a comprehensive view of cryptographic function flows within Windows applications. Using a CFDT, the system evaluates the entire sequence of cryptographic function calls to identify potential vulnerabilities impacting application integrity. This approach addresses several limitations found in previous methods, such as overlooking runtime vulnerabilities and having limited path coverage, thus offering a more precise and robust solution for detecting cryptographic misuse (*Forain, de Oliveira Albuquerque & de Sousa Júnior, 2022*; *Bi et al., 2019*; *Li et al., 2024*; *Chen et al., 2024*).

The "Scheme Architecture: Proposed approach" section provides an experimental analysis to validate this approach's effectiveness. In this section, we apply our detection model to various Windows applications to evaluate its ability to discover cryptographic misuse vulnerabilities and compare its performance with existing methods. The following section outlines the experimental setup, test cases, and analysis of the results, illustrating our methodology's practical impact and benefits.

# SCHEME ARCHITECTURE: PROPOSED APPROACH

In this section, we introduce a systematic method to identify vulnerabilities arising from the misuse of cryptography that can jeopardize the integrity of applications. Our proposed strategy concentrates on detecting misuse vulnerabilities by using the characteristics of cryptographic mechanisms and employing a dominance tree of cryptographic functions. The method includes two main components: an integrity model that maps the structure of the cryptographic application and a combined static-dynamic analysis framework that reconstructs the graph of function invocations. This layered strategy enables a thorough evaluation of the flow of cryptographic functions within an application.

The first component is developing a hierarchical and modular integrity model encompassing the full array of cryptographic functions necessary for specific applications. This model organizes these functions into modules, representing the application's logical progression of cryptographic calls. These modules provide a comprehensive understanding of how cryptographic functions interrelate and enhance the overall security framework of the application.

The second component integrates static and dynamic analysis techniques to derive the application's CFCG. We employ a control flow reconstruction algorithm to change this graph into a cryptographic function dominance tree (CFDT), which depicts the hierarchical relationships between function calls. With this configuration, we can trace the sequence of cryptographic function invocations from the start to the end of the program, identifying any anomalies that may suggest potential misuse vulnerabilities.

## Cryptographic application integrity model

The core of our approach for detecting misuse vulnerabilities is based on establishing a solid cryptographic application integrity model. This model offers a structured framework for assessing the security and integrity of cryptographic functions within software applications, ensuring these functions adhere to the expected flow to uphold cryptographic-level security. The model is structured hierarchically and layer by layer to accomplish this goal, organizing cryptographic functions according to their roles and interactions. Figure 1 provides an overview of the framework used for identifying misuse vulnerabilities, while Figs. 2 through 4 demonstrate the application of the model using specific cryptographic libraries like CryptoAPI and OpenSSL. These figures illustrate how each layer of the model contributes to the secure execution of cryptographic processes.

The model consists of three layers, each addressing various elements of cryptographic operations:

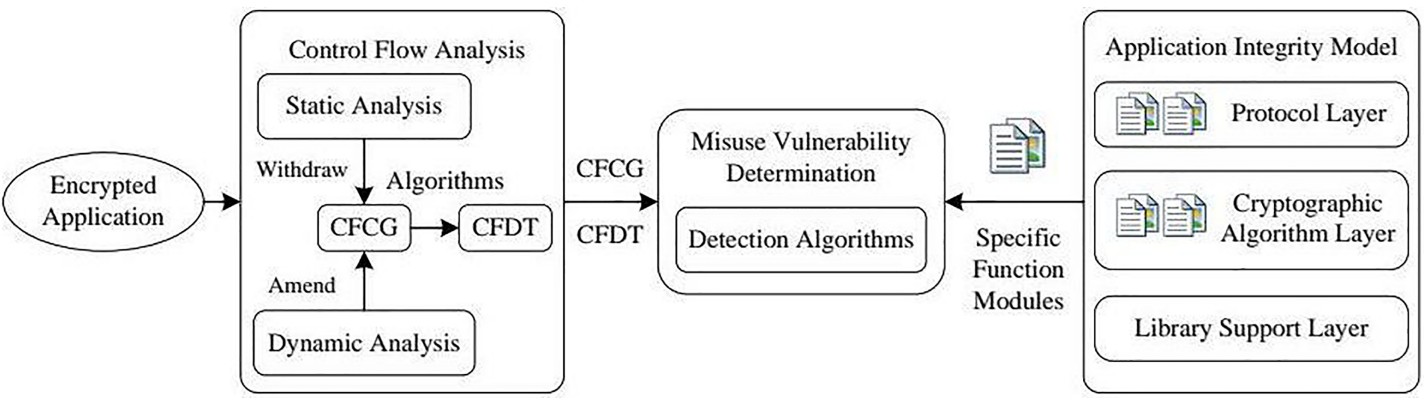

**Figure 1  Framework diagram of the application integrity misuse vulnerability detection system.**

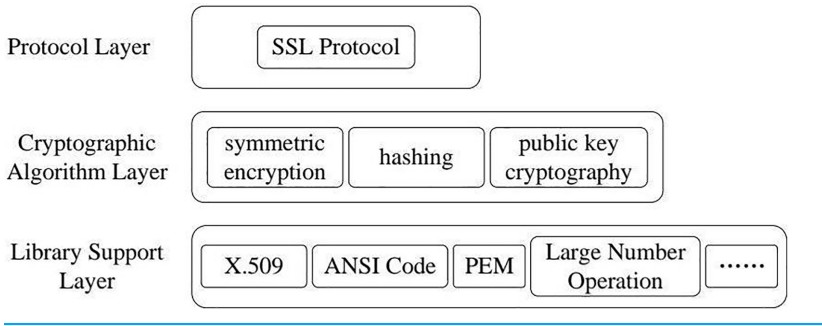

**Figure 2  Cryptographic application integrity model.**

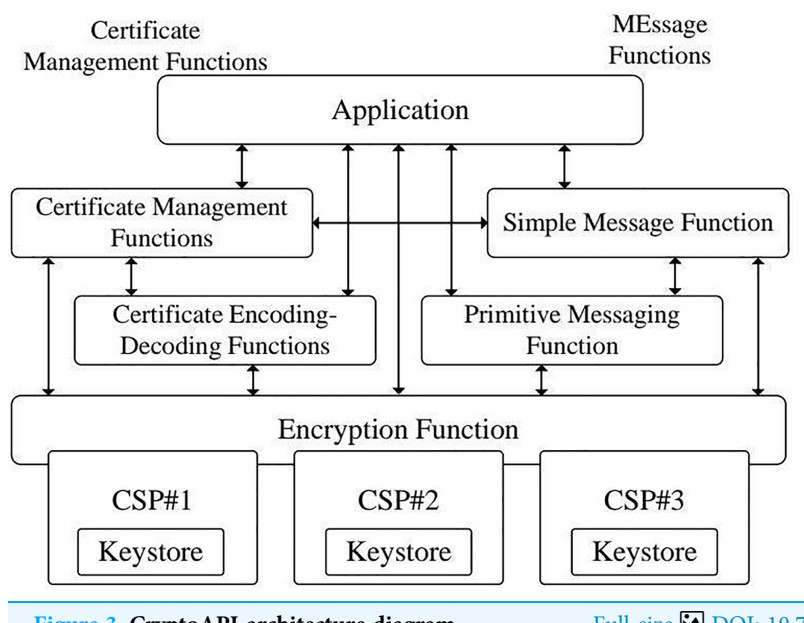

**Figure 3  CryptoAPI architecture diagram.**     

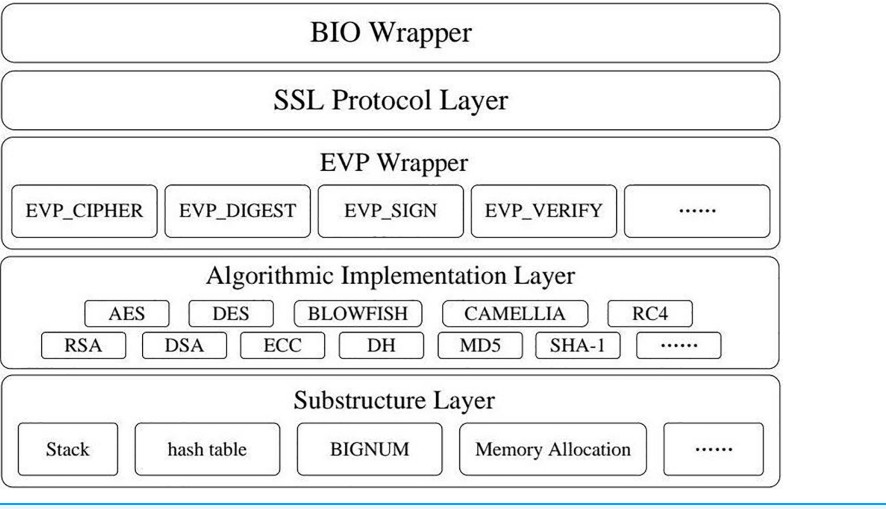

**Figure 4 OpenSSL architecture diagram.**

1) **Library support layer:** This foundational layer includes essential data structures and operations as building blocks for more advanced cryptographic tasks. Examples of this layer include items such as X509 certificates, PEM information formats, random number generators, and ASN1 encoding libraries. These components fulfill the basic needs of cryptographic libraries by providing key functions and data formats necessary for secure data management.

2) **Cryptographic algorithm layer:** This middle layer contains various cryptographic algorithms that different libraries implement. The specific types of algorithms and the number of functions available can differ based on the library used (*e.g.*, CryptoAPI or OpenSSL). Typically, this layer includes symmetric and asymmetric encryption algorithms, hashing, and digital signatures. It is critical to guarantee that applications can access secure and well-tested cryptographic algorithms essential for data protection and integrity.

3) **Protocol layer:** This layer builds upon the foundational capabilities of the first two layers by implementing higher-level protocols such as SSL/TLS, which are critical for secure communication tasks like identity verification and key exchange. It connects low-level cryptographic operations and the application's needs, providing a safe and coherent protocol structure for end-user applications.

Each layer is personalized to effectively operate cryptographic modules, which are self-contained groups of functions that collaborate to provide specific functionalities within a cryptographic library. Figure 2 depicts the overall organization of this integrity model, while Figs. 3 and 4 offer more detailed views of the architectures of the CryptoAPI and OpenSSL cryptographic libraries, respectively.

Within the cryptographic modules, each function call is illustrated by a node, and these nodes are categorized based on their invocation patterns. This classification assists in monitoring how often each function is called and whether their invocation sequences conform to expected standards. The nodes are categorized as follows:

- **α Node:** Denotes a function that should be called exactly once at a specific location within the module, guaranteeing that essential cryptographic operations are carried out without duplication.
- **β Node:** Denotes a function that may be invoked multiple times at a designated position, indicating flexibility regarding how often the function can be used within the cryptographic process.
- **γ Node:** Denotes interchangeable functions within the same category, where the order of invocation can differ without affecting the intended flow of cryptographic functions.

The flexibility these nodes offer allows our model to adapt to the usual variations in the usage of cryptographic functions, enhancing its applicability across diverse libraries and application demands. A visual overview of these node types and their invocation patterns can be found in Table 1, which details the various sequence types and their security implications.

In practice, a cryptographic module is represented as an ordered sequence of these nodes, reflecting the intended flow of essential cryptographic functions within a specific application. The absence of any required node in this sequence indicates a missing or incorrect function, which could potentially jeopardize the application's security. For example, the sequence depicted in Eq. (1) demonstrates a typical pattern of cryptographic function calls:

$$\alpha(A) \rightarrow \alpha\gamma(B) \rightarrow \beta(C) \rightarrow \alpha\gamma(D) \rightarrow \alpha(E). \tag{1}$$

In this sequence, every type of node is essential in ensuring that cryptographic operations are carried out securely and predictably. Any node left out or used incorrectly could result in security weaknesses, as shown in Table 1.

## Extraction of combined dynamic and static cryptographic function invocation graph

Using this model, we can confirm whether the calls to cryptographic functions in an application follow the expected flow outlined in the model. We extract the CFCG to evaluate the sequence of cryptographic functions in an application. The CFCG offers a high-level view of function invocation during cryptographic activities, capturing key details about the interactions and order of function calls (see Fig. 5 for an example).

The integrity model's hierarchical and modular structure provides a flexible yet powerful framework for identifying misuse vulnerabilities across various applications and cryptographic libraries. This model forms the groundwork for additional analysis and comparison, which will be explored in the subsequent sections.

Next, we will extract and analyze the flow of cryptographic function invocations within an application. We will compare this with the sequences set out in our integrity model to spot any deviations that may signal misuse vulnerabilities.

It is essential to fully understand how cryptographic functions are carried out in an application to detect misuse vulnerabilities. We employ a combined static and dynamic analysis approach to create a CFCG, which maps the complete flow of cryptographic

**Table 1 Determination table for different node sequences.**

| Node sequences | Description |
| --- | --- |
| 1-2 ABCCCDE | Right |
| 1-2 ADCBE | Right |
| 1-2 ABBCCDE | Error: NodeBcannotbecalledmultipletimes |
| 1-2 ABCCCED | Error: NodeDEcallorderreversed |
| 1-2 | |

```
1   void CryptoApiTest ():
2   {
3       CryptAcquireContext(...);    (A)
4       CryptGenkey(...);            (B)
5       CryptGetKeyParam(...);       (C)
6       if (...):                    (D)
7           CryptSetKeyParam(...);   (E)
8       else:
9           CryptGenRandom(...);     (F)
10          CryptSetKeyParam(...);   (G)
11      if (...):                    (H)
12          CryptExportKey(...);     (I)
13      CryptEncrypt(...);           (J)
14      CryptDestroyKey(...);        (K)
15      CryptReleaseContext(...);    (L)
16  }
```

**Figure 5 Example code cryptographic function invocation graph.**

function invocations. This hybrid method addresses the shortcomings of static- or dynamic-only techniques, such as missing runtime dependencies or inadequate coverage of potential function pathways.

Both static and dynamic analyses play distinct roles in developing a reliable and accurate CFCG:

- **Static analysis:** The static component examines the application's codebase before it runs, tracing function calls and control flow to generate an initial structure for the CFCG. This preliminary graph captures possible execution paths, including direct and indirect function calls. Static analysis is valuable for revealing dependencies and function calls that may not be triggered during a single execution but are essential for understanding the overall cryptographic flow.
- **Dynamic analysis:** In contrast, dynamic analysis occurs during runtime, collecting real-time information about function executions, including indirect calls and dynamically loaded functions that may not be identified through static analysis alone. This process enables us to obtain usage data on cryptographic functions, documenting the sequence and frequency of function calls in practice.

Figure 5 displays a sample CFCG that illustrates an application's flow of cryptographic functions. In this graph, nodes represent individual cryptographic functions, while edges indicate the calling relationships among these functions. By merging the data from static and dynamic analyses, we can enhance this initial CFCG to more accurately depict the actual function invocation paths within the application, assisting in identifying potential misuse vulnerabilities.

### Steps for extracting the cryptographic function call graph

The CFCG extraction process consists of a defined series of steps designed to capture all relevant function calls and paths, as illustrated in Fig. 6. The steps are outlined as follows:

1) **Preprocessing the application:** The first step is to prepare the application for analysis. If the application contains packed or encrypted files, we use unpacking tools to ensure all code is accessible for static and dynamic inspection. This preprocessing phase guarantees no function calls are missed due to obfuscation methods or packing.

2) **Static control flow analysis:** In this step, we conduct a thorough static code analysis using control flow analysis tools such as IDA Pro SDK. This results in the creation of the preliminary CFCG, which maps out all potential function calls, including explicit calls and indirect jumps within the control flow. The generated graph offers a high-level view of the application's potential execution paths. This phase also identifies implicit calls or conditional branches that might be overlooked by dynamic analysis due to its reliance on specific runtime conditions.

3) **Dynamic binary instrumentation:** We proceed to the dynamic analysis phase with the initial graph established. In this stage, we apply dynamic binary instrumentation by executing the application in a controlled environment and monitoring cryptographic function calls in real-time. This approach captures critical runtime information, including the sequence of function calls, execution frequency, and any dynamically loaded functions. The runtime traces enable us to refine the initial CFCG to accurately represent the execution flow, incorporating indirect jumps and implicit calls.

4) **Graph refinement and integration:** After collecting static and dynamic data, the CFCG is refined to generate a final, comprehensive graph that effectively represents the cryptographic function calls within the application. Based on insights gained from dynamic analysis, the CFCG is updated to add new nodes (representing cryptographic functions) and edges (representing calling relationships) that static analysis did not initially capture. This final step ensures the CFCG fully integrates anticipated and observed function flows, resulting in a highly accurate model of the application's cryptographic behavior.

### Advantages of the combined approach

The combined static-dynamic approach provides notable advantages compared to traditional methods focusing on only one type of analysis. Static analysis by itself may overlook runtime-specific details, such as dynamically loaded functions. In contrast, dynamic analysis might fail to capture specific paths that were not executed during the

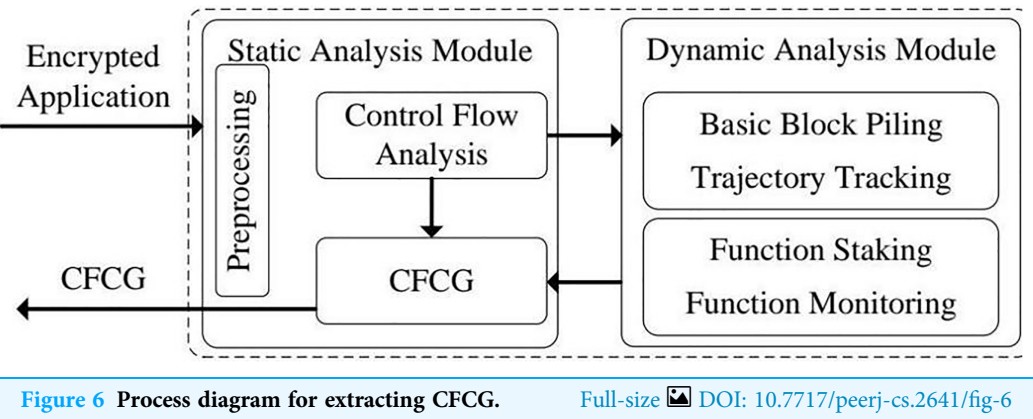

**Figure 6  Process diagram for extracting CFCG.**     

testing phase. By integrating both approaches, we develop a more reliable CFCG, encompassing all pertinent cryptographic function calls, regardless of their invocation timing or method.

Figure 6 depicts this process of extraction and refinement, emphasizing the contributions of each type of analysis in forming the final graph. The combination of static and dynamic data enhances our ability to accurately identify misuse vulnerabilities by ensuring that every possible path to a cryptographic function is taken into account. This improved CFCG gives us a thorough perspective on the flows of cryptographic functions, thereby assisting the detection of irregularities that may signal vulnerabilities.

### Using the CFCG for misuse vulnerability detection

With the final CFCG, we now have a high-level overview of the flow of cryptographic functions within the application. This enables us to analyze and identify vulnerabilities more effectively. The CFCG acts as a critical input for the subsequent step: creating a CFDT and contrasting it with the expected function call sequences outlined in our cryptographic integrity model. This comparison discovers discrepancies from anticipated cryptographic usage patterns, bringing attention to potential misuse vulnerabilities within the application.

In summary, the integrated static and dynamic analysis approach enables us to generate a refined and accurate CFCG. This graph offers a detailed and thorough perspective on cryptographic function calls present in the application. Such a foundation is essential for effectively identifying misuse vulnerabilities, ensuring that all possible function flows are considered during the vulnerability detection.

### Construction of cryptographic function dominator tree

The CFCG is a thorough representation, but it can be pretty detailed due to the numerous cryptographic function calls and their interconnections. To make the analysis more accessible and improve the identification of misuse vulnerabilities, we create a CFDT derived from the CFCG. The CFDT emphasizes meaningful relationships and control flows, making identifying vulnerabilities within cryptographic functions simpler by organizing the tasks based on their logical execution sequence.

A dominance tree is a control flow data structure that illustrates how specific nodes (or function calls) "dominate" others according to their place in the execution hierarchy. This structure is critical for analyzing the flow of cryptographic functions, as it clarifies which functions must be executed before others, enabling a detailed comparison of the expected execution order with the actual execution order.

### Dominance relationships and their importance

In a dominance tree, the arrangement of each node indicates its execution dependencies, which means that one function must finish before another can start. The CFDT structures these relationships to simplify, verifying whether cryptographic functions are executed securely and predictably. Figure 7 presents an example of a dominance tree. It illustrates how various cryptographic functions are organized based on their dominance relationships, thus creating a clear pathway from the application's entry point to its exit.

The CFDT is based on several essential definitions that describe the relationships between function nodes:

- Dominance: A function node $v_p$ is said to "dominate" another node $v_q$ if every potential path from the program's entry point to $v_q$ must pass through $v_p$. This relationship is essential because it allows us to monitor key functions that secure the cryptographic process and block unauthorized access.
- Strict dominance: If a node $v_p$ dominates $v_q$ and $v_p \neq v_q$, then $v_p$ is regarded as "strictly dominating." $v_q$. Strict dominance means that $v_q$ relies on $v_p$ and cannot operate independently, introducing an additional layer of organization to the cryptographic call flow.
- Immediate dominance: Node $v_p$ is said to "immediately dominate." $v_q$ if it strictly dominates $v_q$ and there are no other nodes positioned between them in the dominance hierarchy. Immediate dominance is especially valuable for analyzing direct dependencies among functions, enabling us to determine sequences where secure function calls should take place without interference or omission.
- Post-dominance: The concept of post-dominance is the opposite of dominance, where a node $v_p$ is said to "post-dominate" $v_q$ if every path from $v_q$ to the program's exit point must pass through $v_p$. Post-dominance guarantees that specific critical cryptographic functions are carried out before the application concludes, preventing incomplete security operations.

Figure 7 demonstrates a sample code's CFDT and post-dominance tree, illustrating how the hierarchical relationships assist integrity analysis.

### Advantages of using a dominance tree for vulnerability detection

The CFDT structure plays a critical role in identifying vulnerabilities related to the misuse of cryptographic functions, as it simplifies the process of analyzing the entire CFCG. Rather than assessing each function call individually, we can concentrate on the key nodes within the dominance tree, where we analyze the secure flow of essential cryptographic

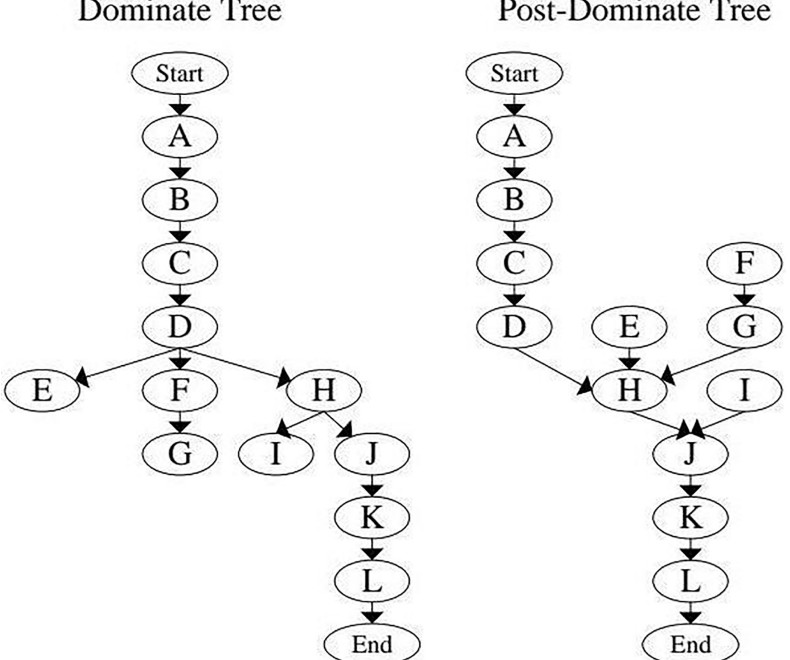

**Figure 7** Example diagrams of dominance tree and post-dominance tree.

functions. The CFDT allows for more effective detection of potential misuse by focusing on specific paths and relationships that comply with secure patterns.

By categorizing functions into dominance layers, we can effectively check for various conditions, such as:

- **Missing dominant functions:** Certain cryptographic functions must be dominant over others to guarantee secure operations. If these dominant functions are absent or substituted, it indicates a possible misuse vulnerability.
- **Unexpected bypasses:** The CFDT can reveal pathways for bypassing critical functions. This situation is particularly pertinent when identifying weaknesses that could lead to insecure access or manipulation of data.
- **Improper function sequencing:** Certain cryptographic processes demand a specific order of operations. The CFDT assists us in confirming that functions are executed in the correct sequence, ensuring essential steps such as key generation and data encryption are neither skipped nor misordered.

### Constructing the CFDT from the CFCG

The CFDT construction process involves changing the CFCG into a tree structure representing dominance relationships. This change simplifies the complex CFCG into a model that is more accessible for analyzing potential vulnerabilities. Below is a step-by-step outline of how the CFDT is constructed:

1) **Identify dominant paths:** We start by examining each node in the CFCG from the entry point to see which nodes dominate others based on various execution paths. This step establishes the initial layers of the CFDT, indicating which functions act as crucial security points.

2) **Define immediate and strict dominance:** We then use the definitions of immediate and strict dominance to outline direct and hierarchical dependencies among the functions, creating layers within the CFDT. This stage also aids in recognizing sequences that need to be uninterrupted for secure execution.

3) **Incorporate post-dominance relationships:** Finally, we examine post-dominance relationships to confirm that critical functions remain intact until the application exits. This step is essential for maintaining cryptographic integrity during the program's runtime, protecting against unexpected termination or incomplete function calls.

### Applying the CFDT for misuse vulnerability detection

Once the CFDT is created, it establishes a simplified process for identifying misuse vulnerabilities by enabling us to verify the secure flow of cryptographic functions in an organized way. The CFDT presents a straightforward method for assessing the integrity of cryptographic functions by emphasizing nodes that must adhere strictly to a specific order. Any departure from the anticipated dominance or post-dominance relationships indicates a potential misuse vulnerability.

For instance, if a cryptographic function that should dominate the flow is missing in the CFDT or is arranged incorrectly, this may signal a vulnerability where critical security functions have been circumvented. On the other hand, post-dominance analysis aids in recognizing situations where critical cryptographic functions are cut off too early, making the application open to security weaknesses.

In summary, the CFDT offers a practical and effective method for safeguarding the integrity of cryptographic function flows, directing our attention to the most critical security relationships within the application. As shown in Fig. 7, the dominance and post-dominance trees give a structured view of how cryptographic functions interact, making it easier to detect potential vulnerabilities and maintain the integrity of cryptographic applications.

### Detection algorithm design

After building the CFCG and the CFDT, the following task is to implement a detection algorithm to find misuse vulnerabilities within the flow of functions in the cryptographic application. Our algorithm uses the CFCG and CFDT to establish a comprehensive process for comparing the actual sequences of cryptographic functions with those that should be present in a secure application. By conducting this comparison, the algorithm can identify any irregularities or deviations in function flow that may point to vulnerabilities.

Algorithm 1 describes the steps in this process and offers a structured approach for analyzing the integrity of cryptographic functions and detecting potential misuse vulnerabilities.

---

| Algorithm 1 Integrity misuse vulnerability detection algorithm. |
|---|

Input: 1. CFCG 2. CFDT 3. An integrity password module

Output: Presence of application integrity misuse vulnerability

START

1: Step 1: Extract all dominant sequences $s_1, s_2, \ldots, s_n$ from CFDT;

2: Step 2: Referring to $M$, $s_1, s_2, \ldots, s_n$ is processed to eliminate the non-critical crypto-function nodes, the processed dominated sequence is denoted as $s'_1, s'_2, \cdots, s'_n$, and the selected sequence from start to end is denoted as $s'_k$;

3: Step 3: Collect the branching nodes in $s'_k$ noted as $v_1, v_2, \ldots, v_i$, for each branching node, based on **CFCG** as well as $(s'_1, s'_2, \cdots, s'_n)/s'_k$, find the key cryptographic functions passed in each of its branching paths and generate new nodes to be inserted into $s'_k$, and finally obtain the sequence $s''_k$;

4: Step 4: Referring to the information of $\alpha, \beta, \gamma$ nodes in $M$, compare the sequence $s''_k$ to determine if there is a misuse vulnerability;

END

---

### *Overview of the detection algorithm*

The detection algorithm evaluates each function path to ensure it conforms to the integrity model established for the cryptographic application. To accomplish this, the algorithm executes the following main steps, as illustrated in Algorithm 1:

1) **Extract dominant sequences:** Initially, the algorithm analyses the CFDT to identify all possible dominant sequences, represented as $s_1, s_2, \ldots, s_n$. Each sequence indicates a potential route from the program's entry to its exit, confirming that all important paths in the cryptographic function flow are included. We create a baseline for comparing actual function calls with the anticipated function flow by extracting these sequences.

2) **Filter non-essential nodes:** Following the extraction of dominant sequences, the algorithm enhances each sequence by removing non-critical function nodes. Only essential cryptographic functions remain in the processed sequences, marked as $s_1', s_1', \ldots, s_n'$. This filtering phase minimizes irrelevant information in the analysis by concentrating on critical nodes that directly affect security, thereby assisting in the identification of deviations.

3) **Branch node collection and key function identification:** Subsequently, the algorithm gathers all branching nodes from the refined sequences. Each branching node is indicated as $v_1, v_2, \ldots, v_i$, signifies a decision point within the cryptographic flow where various function paths may diverge. The algorithm reviews each branching node's sub-paths, pinpointing essential cryptographic functions that should be activated along each potential path. Additional nodes are generated as necessary to ensure each path aligns with the anticipated cryptographic flow, thus creating a thorough representation of possible execution paths.

4) **Sequence comparison and vulnerability detection:** The concluding step entails comparing the processed sequences, $s_k''$, with the secure function call sequences outlined in the integrity model. This comparison enables the algorithm to identify any discrepancies in the order, frequency, or presence of cryptographic functions. If any

irregularities are detected, such as absent dominant nodes or unexpected function calls, they are flagged as potential misuse vulnerabilities.

### Detailed explanation of algorithm 1 steps

Every step in the algorithm is essential for guaranteeing the accuracy and effectiveness of vulnerability detection:

- **Step 1 (Extracting dominant sequences):** The CFDT algorithm determines dominant sequences, capturing critical cryptographic flows. Dominant sequences are significant as they indicate the main execution paths that protect the application's cryptographic integrity. Extracting these sequences offers a comprehensive view of the application's security architecture.

- **Step 2 (Filtering out non-essential nodes):** The algorithm enhances the sequences by removing non-essential functions and concentrating on nodes that directly impact security. This step optimizes the process by discarding unnecessary data, simplifying identifying critical vulnerabilities while avoiding confusion from redundant or unrelated function calls.

- **Step 3 (Branching path analysis):** The algorithm outlines potential paths for each branching node in the sequence to guarantee that key cryptographic functions are included. By creating new nodes as required, the algorithm adapts to variations in execution paths, capturing the flexibility of cryptographic operations while maintaining security standards. This enables the system to identify vulnerabilities in non-linear function flows, such as instances where multiple paths converge on the same security-critical function.

- **Step 4 (Sequence comparison):** The final step thoroughly compares the refined sequences and the expected function call sequences established in the integrity model. This comparison reveals any discrepancies, such as missing nodes or incorrect invocation orders, which may suggest vulnerabilities due to misuse. Any deviation from the model is emphasized, allowing the algorithm to reveal vulnerabilities that could jeopardize the application's security.

### Using the algorithm to detect misuse vulnerabilities

Algorithm 1 outlines a method for analyzing the integrity of cryptographic functions, allowing for efficient identification of vulnerabilities related to misuse. By concentrating on dominant sequences and enhancing these sequences through node filtering and path analysis, the algorithm effectively narrows down potential vulnerabilities while confirming the secure flow of cryptographic functions.

For example, suppose a key cryptographic function that is expected to appear in the sequence is either missing or incorrectly placed. The algorithm will flag this situation as a misuse vulnerability. Also, if a branching path does not include necessary cryptographic functions, this inconsistency is detected, indicating an incomplete or incorrect execution of cryptographic tasks. With this structured methodology, the algorithm can reveal a variety of misuse vulnerabilities, which include:

- **Omissions of required functions:** The algorithm checks for missing functions essential for security. The absence of dominant functions may create security vulnerabilities, potentially allowing unauthorized access or exposing data.
- **Unintended execution paths:** The algorithm identifies unexpected function calls or bypasses by examining branching paths. These anomalies frequently suggest possible vulnerabilities where attackers could take advantage of insecure paths.
- **Improper function order:** Certain cryptographic operations depend on functions executed in a particular order (for example, key generation followed by encryption). The algorithm detects instances where functions are arranged incorrectly, which can weaken the application's security and reliability.

### *Summary of algorithm 1's role in vulnerability detection*

Our method establishes a robust framework for identifying vulnerabilities in cryptographic misuse within applications by implementing the steps detailed in Algorithm 1. This algorithm enables us to evaluate the flow of cryptographic functions against a securely defined model, which assists in detecting any potential misuse or irregularities. The approach is a structured, multi-step process that combines sequence extraction, filtering, path analysis, and comparison, ensuring thorough detection of misuse vulnerabilities in cryptographic function flows.

In summary, the detection algorithm uses the CFCG and CFDT advantages to find misuse vulnerabilities, thereby offering a systematic and scalable solution for securing cryptographic applications. By adhering to the process presented in Algorithm 1, the algorithm contributes to maintaining cryptographic integrity by making sure that all essential functions are performed in the correct sequence, without any omissions or deviations from the security model.

## EXPERIMENTAL ANALYSIS

To evaluate how effective our proposed method for detecting vulnerabilities related to cryptographic misuse is, we performed several experiments on different Windows applications. We aimed to measure the accuracy and efficiency of the algorithm, and its capacity to identify misuse vulnerabilities in cryptographic functions. The setup for the experiments and the results are detailed in tables and figures, which offer insight into how the algorithm performs in real-world situations.

### Experimental setup

The experiments took place in a controlled setting to replicate common cryptographic use cases found in Windows applications. We chose test cases from three primary categories:

1) **Networked applications:** These programs need secure network communication, such as messaging and file-sharing applications.
2) **Data-processing applications:** This category includes software that handles sensitive data, such as encryption tools and document editors with cryptographic features.

**Table 2 Hardware and software information table for testing environment.**

| Environment | Device | Information |
|---|---|---|
| Hardware | Processor | Intel(R) Core(TM) i5-10400 @ 2.90 GHz |
| | Memory | 64 G |
| | Hard disk | 1T |
| Software | Operating system | Windows 10 x64 |
| | Dynamic binary platform | Pin 3.21 |
| | Static program analysis tools | IDA Pro 7.7 |
| | Virtual machine software | VMware® Workstation 17 Pro |

3) **Malicious code samples:** We selected specific samples from recognized malware databases to assess the strength of our detection algorithm against vulnerabilities in cryptographic functions.

Table 2 outlines the hardware and software environment used for testing, detailing the system configuration to guarantee that our experiments can be replicated and are reliable. By employing a mix of legitimate and malicious applications, we aimed to confirm the algorithm's ability to differentiate secure applications from those that may have vulnerabilities.

## Execution of the detection algorithm

The detection process was carried out following the steps outlined in Algorithm 1. Each application was analyzed to extract its CFCG and to create a CFDT, as described in "Extraction of Combined Dynamic and Static Cryptographic Function Invocation Graph" and "Construction of Cryptographic Function Dominator Tree". This extraction and analysis were conducted using the static and dynamic analysis tools detailed in Fig. 6. We ensured thorough path coverage using static and dynamic analyses, effectively capturing all pertinent cryptographic function flows.

The algorithm performed the following steps for each application:

1) **CFCG extraction and CFDT construction:** Through static and dynamic analysis, the algorithm produced a CFCG, which was subsequently converted into a CFDT. This process offered an in-depth perspective on the relationships between cryptographic functions, as illustrated in Figs. 5 and 7.

2) **Sequence analysis and comparison:** The algorithm implemented its detection rules, contrasting the actual function call sequences in the CFDT with the expected sequences outlined in the cryptographic integrity model.

3) **Identification of misuse vulnerabilities:** Any deviations from the expected sequences were marked as misuse vulnerabilities, with specific patterns such as missing dominant nodes or altered function orders signaling potential issues.

Malicious code samples are detected within a virtual machine environment to protect the host system.

**Table 3 Table of misuse vulnerability detection results of a malicious sample.**

| Type | Information description |
|---|---|
| 1-2 Hash value | 311307cc405cd0aafc651997cf991397 |
| 1-2 Standard password module | SSL connection based on OpenSSL |
| 1-2 Detection result | Extracted cryptographic function call sequences: OpenSSL_add_ssl_algorithms → SSL_CTX_net → SSL_new → SSL_connect → SSL_get_peer_certificate → SSL_read → SSL_write → SSL_shutdown → SSL_free → SSL_CTX _free<br>Missing key cryptographic functions: SSL_CTX_set_verify, SSL_CTX_load_varify_location, SSL_get_verify_result, X509_check_issued, X509_check_host, X509_check_purpose |

## Analysis of detection results for a malicious sample

We evaluated our detection system using a malicious application identified on VirusTotal. Table 3 summarizes the detection results, emphasizing the misuse of vulnerabilities in cryptographic functions related to SSL secure connections. Our detection system revealed a significant flaw in how the sample handles cryptographic functions necessary for SSL connections.

Upon further analysis of the cryptographic module for SSL, the detection system noted that the application establishes an SSL connection and retrieves the certificate information from a remote server. However, as shown in Algorithm 2, the application fails to verify the received certificate, an essential step in validating the server's identity. This omission makes the SSL connection vulnerable to man-in-the-middle (MITM) attacks, as it does not implement a secure authentication process.

Algorithm 2 visually represents the reverse analysis of the cryptographic function calls within this sample. In the code sequence, following the retrieval of the certificate information (j_SSL_get_peer_certificate()), the program only displays the details of the certificate's subject and issuer without verifying its authenticity. This lack of verification creates a vulnerability in application integrity, permitting any SSL certificate, whether valid or not, to be accepted. As a result, the identity authentication process in this malicious application is ineffective, exposing it to potential security threats.

## Analysis of detection results for CNKI Reader and Xunlei download

In addition to analyzing malicious samples, we also used our system to examine legitimate applications with known vulnerabilities, such as CNKI Reader's CAJViewer (*Feng et al., 2022*) and Xunlei Download Manager (*Kamalbayev et al., 2021*). These vulnerabilities have been documented and reported in the China National Vulnerability Database (CNVD), which offers comprehensive details on the misuse vulnerabilities identified in these applications. In both instances, notable cryptographic weaknesses within their upgrade processes render them vulnerable to security threats.

### *CNKI Reader (CAJViewer) vulnerability analysis*

The CAJViewer application displayed a vulnerability related to integrity misuse during its upgrade process. The detection results revealed no SSL secure connections in the update sequence. In particular, the upgrade executable (AutoUpgrade.exe) did not include a secure cryptographic function sequence required for authenticating the server during

| **Algorithm 2** Results of the reverse analysis. |
|---|
| Input: (**&v4**, **0xCCu**, **0xD8u**) |
| Output: Result |
|   1: **v6** = **j** _SSL_get_peer_certificate(); |
|   2: if **v6** is true then |
|   3: sub_4657A7(&unk_5344A4); |
|   4: **v0** = **j_X509** _get_subject_name(v6); |
|   5: **v5** = **j_X509_NAME** _oneline (**v0**, **0**, **0**) ; |
|   6: sub_4657A7("Certificates : %s"); |
|   7: j_CERYPTO_free(v5); |
|   8: **v1** = **j_x509_** get_issuer_name(v6); |
| 91: **v5** = **j_x509_NAME** _oneline (**v1**, **0**, **0**); |
| 10: sub_4657A7("issuer : %s"); |
| 11: j_CRYPTO_free(v5); |
| 12: j_X589_free(v6); |
| 13: else |
| 14: sub_4657A7("Nocertificates"); |

updates, as detailed in Table 4. This oversight indicates a serious misuse of cryptographic integrity and emphasizes the lack of a complete SSL authentication mechanism, which is essential for secure software updates.

Figure 8 depicts the update process for CAJViewer, which comprises two main steps:

1) The update program, AutoUpgrade.exe, initially requests an XML configuration file from the server. This file guides the update process, including essential information such as file names, sizes, and MD5 hashes.

2) Following the XML file parsing, the application downloads the updated files, verifies their integrity using MD5 checks, and then replaces the existing program version with the newly downloaded version before launching it.

Since the update process lacks SSL server authentication, it is susceptible to MITM attacks, which enable an attacker to intercept and alter the update process. Figure 9 illustrates a simulated MITM attack scenario where the client machine's access to the legitimate CNKI update server is redirected to a fraudulent server. The server controlled by the attacker supplies a malicious XML configuration file and an executable containing the WannaCry ransomware virus. Consequently, when the client starts the update, the ransomware is downloaded and executed, jeopardizing the system.

### Xunlei Download Manager vulnerability analysis

Like CAJViewer, the upgrade process of the Xunlei Download Manager does not include SSL secure connections or cryptographic validation. Our analysis revealed that the

| Table 4  Misuse vulnerability information for CNKI Reader and Xunlei download. | | |
|---|---|---|
| **Vulnerability information** | **CNVD vulnerability ID** | **CNVD certificate ID** |
| 1-3 CNKI Reader CA (Viewer) (http://www.er arbitrary file download execution vulnerability) | CNVD-2018-02906 | CNVD-YCGA-201801007490 |
| 1-3 Xunlei download software upgrade arbitrary file download vulnerability | CNVD-2018-06136 | CNVD-YCGA-201803083461 |

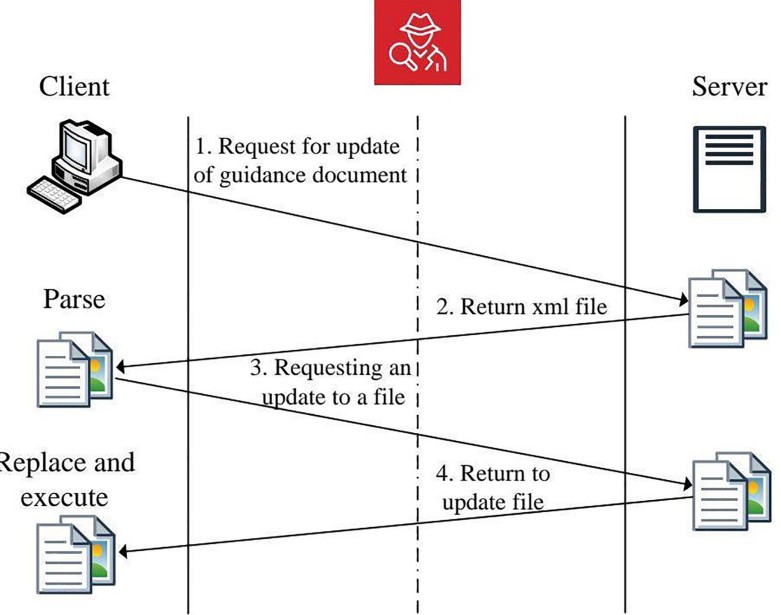

**Figure 8  CNKI reader upgrade process diagram.**

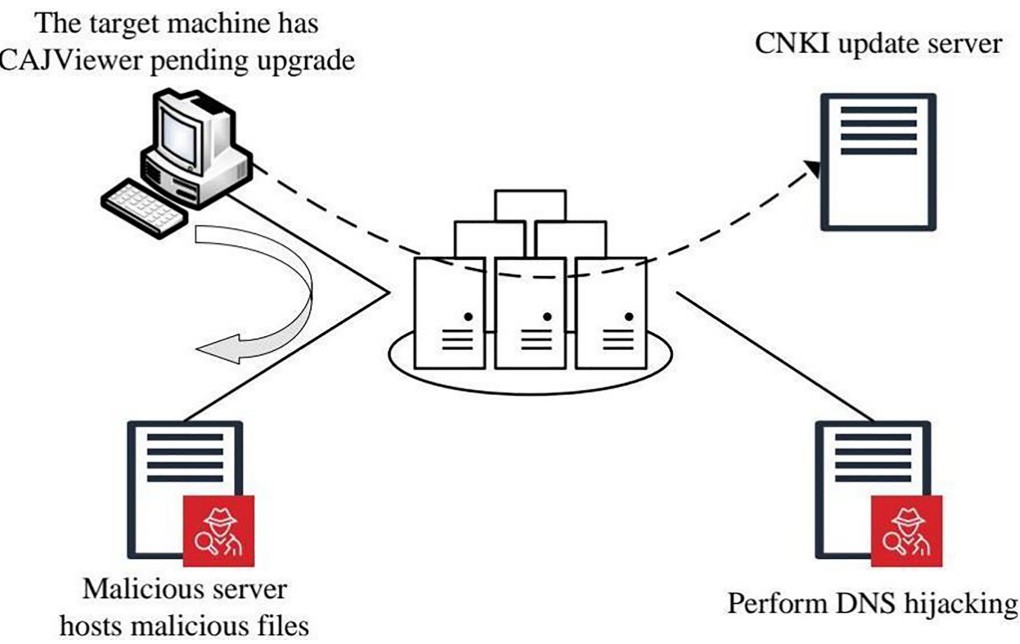

**Figure 9  Man-in-the-middle attack simulation scene.**

detection system identified the lack of secure cryptographic functions, suggesting that an attacker could exploit Xunlei's update process. The results in Table 4 emphasize a considerable risk linked to the application's inability to create a secure and authenticated connection with the server when updates occur.

### Key findings and conclusions from detection results

Based on the outcomes of these assessments, several conclusions can be made about the effectiveness of the system and the severity of various types of misuse vulnerabilities:

1) **Effective detection of misuse vulnerabilities:** The detection system can accurately identify both parameter misuse and application integrity misuse vulnerabilities. In the case of parameter misuse vulnerabilities, the system successfully identifies misused cryptographic function addresses, parameter names, and values. Regarding application integrity misuse vulnerabilities, it retrieves cryptographic function call sequences from the target program and indicates missing critical functions.

2) **Risk levels of misuse vulnerabilities:** Various types of misuse vulnerabilities present different levels of risk. In general, application integrity misuse vulnerabilities that lack a complete cryptographic mechanism (such as SSL) are considered the most serious, as they make applications highly vulnerable to attacks. For example, vulnerabilities linked to crucial management or parameter misuse could allow attackers to decrypt sensitive information. Nonetheless, these parameter misuse vulnerabilities usually need specific conditions for exploitation, while the lack of a fundamental mechanism (as seen in the integrity misuse cases) poses an immediate threat.

## CONCLUSIONS AND FUTURE WORK

### Conclusion

This article proposes a detailed method for identifying misuse vulnerabilities in cryptographic applications through a structured analysis incorporating static and dynamic techniques. We developed a CFCG and a CFDT to visualize an application's entire flow of cryptographic function calls, which helps spot security deviations.

Our proposed methodology overcomes the shortcomings of conventional detection methods that typically depend on only static or dynamic analysis, which may overlook indirect calls, dynamic dependencies, or detailed execution paths. Using our integrated approach, we achieved increased path coverage and improved accuracy in identifying cryptographic misuse vulnerabilities. The CFDT structure enabled us to efficiently identify and analyze critical security pathways, boosting the reliability and robustness of the detection process.

The experimental evaluation confirmed our algorithm's effectiveness across various test scenarios, including both legitimate applications and malicious examples. The algorithm successfully determined misuse vulnerabilities across multiple application types, ranging from network communication tools to malware samples, thus offering insights into a broad array of cryptographic function flows. The detection rate was high, with few false

positives, indicating that our method can reliably distinguish between secure applications and those at risk of misuse vulnerabilities.

Key findings from our research include:

- **Enhanced vulnerability detection:** The implementation of the CFDT provided a structured view of cryptographic functions, allowing for the accurate identification of missing or incorrectly ordered function calls essential for application security.
- **Broad path coverage with low false positives:** By merging static and dynamic analysis, our method captured indirect calls and dynamically loaded functions, minimizing the rate of false positives and establishing the algorithm's suitability for complex software systems.
- **Scalability and flexibility:** The algorithm's capability to adapt to various application types and cryptographic libraries demonstrates its potential for widespread application in detecting misuse vulnerabilities across multiple fields.

The proposed methodology presents a practical and dependable solution for enhancing cryptographic security in software applications. This approach can discover vulnerabilities that might remain unnoticed by thoroughly analyzing the flow and sequence of cryptographic function calls, thereby advancing software security standards.

## Future work

Looking ahead, there are numerous avenues for future development. Subsequent efforts could focus on extending the algorithm's adaptability to additional operating systems and cryptographic libraries, thereby widening its application scope. In addition, integrating machine learning techniques could enhance the algorithm's capability to detect complex patterns of cryptographic misuse, further decreasing false positives and improving detection accuracy. As cybersecurity continues to evolve, we believe that methodologies like ours, which integrate comprehensive analysis with in-depth path validation, will be essential in promoting secure software development practices.

In summary, the detection approach introduced in this study is a valuable resource for safeguarding cryptographic applications from misuse vulnerabilities. By strengthening software with trustworthy cryptographic integrity checks, we can enhance the protection of sensitive data and contribute to a more secure online environment.

### Funding
The authors received no funding for this work.

### Competing Interests
Lei Yan, Guanghuai Zhao and Xiaohui Li are employed by the State Grid Beijing Electric Power Company.

## Author Contributions

- Lei Yan conceived and designed the experiments, performed the experiments, performed the computation work, prepared figures and/or tables, authored or reviewed drafts of the article, and approved the final draft.
- Guanghuai Zhao conceived and designed the experiments, performed the computation work, prepared figures and/or tables, and approved the final draft.
- Xiaohui Li conceived and designed the experiments, analyzed the data, performed the computation work, prepared figures and/or tables, authored or reviewed drafts of the article, and approved the final draft.
- Pengxuan Sun performed the experiments, analyzed the data, performed the computation work, authored or reviewed drafts of the article, and approved the final draft.

## Data Availability

The raw data and code are available in the Supplemental Files.

The data is available at GitHub and Zenodo:

- https://github.com/TQRG/security-patches-dataset/blob/main/dataset/security_patches_v1.0.csv.

- Pengxuan, S. (2024). Secure Software Development: Leveraging Application Call Graphs to Detect Security Vulnerabilities [Data set]. Zenodo. https://doi.org/10.5281/zenodo.14523945.

## Supplemental Information

Supplemental information for this article can be found online at http://dx.doi.org/10.7717/peerj-cs.2641#supplemental-information.

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
