# Peer review of "Secure software development: leveraging application call graphs to detect security vulnerabilities"

_PeerJ Computer Science, doi:10.7717/peerj-cs.2641_

## Round 0.1 · original submission · Major Revisions

Please see both reviewer's detailed comments. The reviews recommend key improvements for clarity and structure. They suggest revising the abstract to include specific outcomes and reorganizing the introduction to better present background, problems, and contributions. The related work section needs recent studies and a summary of findings. Reviewers advise refining figures, tables, and technical terms for readability, adding an experimental setup, and clarifying jargon. The conclusion should discuss limitations, future work, and performance impact. Overall, they recommend enhancing transitions, simplifying complex sections, and correcting typos to improve writing quality.

Reviewer 1 ·

Basic reporting

The idea presented in this paper “Secure software development: Leveraging application call graphs to detect security vulnerabilities” is good. The authors in this paper attempted to develop a dynamic and static vulnerability detection method based on a cryptographic function dominance tree to reduce the risk of compromising the integrity of cryptographic applications. The authors are suggested to address the following comments while revising the paper.

Revise the abstract. Currently, the abstract provides why and what is done. The key findings on the effectiveness of the method developed are not reported.

The introduction and writing of the paper needed to be improved and all the references needed to be cited properly. The current introduction is missing both organization and proper citation. The authors are suggested to improve the introduction section with a proper flow from the background, problem, existing solutions, and research gaps, and lastly research contributions of this study.

Improve and extend related work section. It contains only two big paragraphs. It is suggested that more recent studies be added and organized in a better way. Also, summarize the related work at the end to report the key takeaways from the literature.

Move the section 3 paper structure as a paragraph at the end of the introduction section. A separate section is not needed for this.

Experimental design

Rename “4. Scheme Architecture” to the proposed architecture and methodology.

Algorithm 1 “Integrity Misuse Vulnerability Detection Algorithm” is going out of margin.

Line 243 “…clear represen- of…” Correct the typo and review the paper carefully for all typos.

Before Section 5. Experimental Analysis, add a section or details on “Experimental Setup” that details the environment and configuration of the system, the data or scenario (Malicious sample, CNKI Reader and Xunlei Download) used for evaluating the proposed architecture along with the evaluation criteria or metrics used in this study to report the results.

Validity of the findings

A more detailed discussion on the analysis is required. Also reported and the implications of architecture developed in this study.

Revise the conclusion and identify the limitations and potential future work of this study.

Overall, the writing of the paper also needs improvement.

·

Basic reporting

Overall the manuscript seems to be good but I suggest the following changes to be considered

Abstract lacks specific details about outcomes.
Lack of clear distinction between theoretical and implementation security.
Clarify “cryptographic misuse vulnerabilities” and their types.
Typo in “churp-misuse vulnerabilities.”
Improve transition from background to methodology.
Related Work section could be organized with subheadings.
Section 3 lacks context on improvements over existing methods.
Improve the transition between Sections 3 and 4.
Jargon like “CFCG” and “CFDT” needs consistent definitions.
Cryptographic model structure lacks explanation of layer prioritization.
Figures need more detailed captions.
Figure 5 lacks context for “pseudo-code segment.”
Simplify technical details in CFDT and CFCG extraction for general readers.
Algorithm 1 needs step-by-step explanations of symbols.
Explain “control flow reconstruction” in Section 4.2.
Provide an example of detecting a misuse vulnerability.
Add a summary table for experimental results.
Describe criteria for test case selection in Experimental Analysis.
Clarify terms “dominance sequence” and “dominance relationship” with examples.
“Dominance relationship chain” in Section 4.3 lacks context.
Add detail on CNKI Reader vulnerabilities.
Table 3 needs further explanation.
Define “integrity misuse vulnerability.”
Minor typos in Section 5.2.
Mention performance impact of the proposed method.
Conclusions lack insights on future work.
Discuss limitations, such as false positives.

Experimental design

See "Basic reporting".

Validity of the findings

See "Basic reporting".

Additional comments

No comments.

---

## Round 0.2 · accepted · Accept

Both reviewers have confirmed that their comments are addressed.

Reviewer 1 ·

Basic reporting

no comment

Experimental design

no comment

Validity of the findings

no comment

·

Basic reporting

The authors have adequately addressed all the issues that I raised in my previous report.

Experimental design

The authors have adequately addressed all the issues that I raised in my previous report.

Validity of the findings

The authors have adequately addressed all the issues that I raised in my previous report.

Additional comments

The authors have adequately addressed all the issues that I raised in my previous report.